# Understanding high pressure molecular hydrogen with a hierarchical machine-learned potential

Hongxiang Zong [1,2✉], Heather Wiebe[1] & Graeme J. Ackland [1✉]

The hydrogen phase diagram has several unusual features which are well reproduced by density functional calculations. Unfortunately, these calculations do not provide good physical insights into why those features occur. Here, we present a fast interatomic potential, which reproduces the molecular hydrogen phases: orientationally disordered Phase I; broken-symmetry Phase II and reentrant melt curve. The $H_2$ vibrational frequency drops at high pressure because of increased coupling between neighbouring molecules, not bond weakening. Liquid $H_2$ is denser than coexisting close-packed solid at high pressure because the favored molecular orientation switches from quadrupole-energy-minimizing to steric-repulsion-minimizing. The latter allows molecules to get closer together, without the atoms getting closer, but cannot be achieved within in a close-packed layer due to frustration. A similar effect causes negative thermal expansion. At high pressure, rotation is hindered in Phase I, such that it cannot be regarded as a molecular rotor phase.

[1] Centre for Science at Extreme Conditions and School of Physics and Astronomy, University of Edinburgh, Edinburgh EH9 3ET, UK. [2] State Key Laboratory for Mechanical Behavior of Materials, Xi'an Jiaotong University, Xi'an 710049 Shaanxi, China. ✉email: zonghust@mail.xjtu.edu.cn; gjackland@ed.ac.uk

Since the discovery of solid molecular hydrogen in 1899, the nature of this phase has remained controversial[1]. It is now believed that the solid "Phase I" comprises rotating hydrogen molecules on a hexagonal close-packed lattice[2]. With increasing pressure the rotation becomes hindered[3] by inter-molecular interactions, both steric and electrostatic, leading ultimately to phase transformations to a low temperature Phase II[4], in which quadrupole–quadrupole interactions (EQQ) arrest the rotation[5], and a high-pressure Phase III[6,7], in which steric interactions dominate.

Experimental study of these phases has proved challenging. X-ray study showed the hcp structure, but could not resolve molecular orientation at low temperature[8], and the first room temperature only completed in 2019[9]. Raman spectroscopy shows peaks corresponding to quantum rotors at low pressure, which gradually broaden and shift with pressure, and a distinctive sharp phonon mode which rules out cubic close packing as a structure[10–15]. The melt line has a strongly positive Clapeyron slope at low pressures, with a turnover around 100 GPa[16–19]. The negative slope means that even though the solid is hexagonal "close-packed", the liquid must be even denser. The turnover also means the liquid has higher compressibility, but how this comes about remains unexplained. X-ray studies at low temperature traversing Phase I–II–III do not show any convincing structural changes, in part because it has proven impossible to get sufficient resolution to determine the molecular orientation[8,9].

Spectroscopy gives vibrational data, which are still insufficient to determine the structures of phases II, III, and IV. There have been many and varied attempts to identify the structures via simulations[20–30]. However, a consensus has not yet been reached. Based on fully ab initio calculations, including density functional theory (DFT) or quantum Monte Carlo (QMC)[31], a number of candidate structures have been proposed for Phase II and III. Besides differing molecular orientation, they are all similar, consisting of primitive cells with lattice sites close to hcp[22]. Among the structures, the $P2_1/c$-24, $C2/c$-24, and $Pc$-48 structures provide low-energy candidate structures for phases II, III, and IV.

The modern theory of the structure of these phases is based around electronic structure calculations. The early work involved calculating the ground state, assuming classical nuclei, then adding quantum-nuclear effects via the quasiharmonic approximation. This methodology, whether based on DFT or QMC, predicts hcp-like ground states for Phases I–III in agreement with X-ray data. However the spectroscopic signature of the Phase II—the appearance of many sharp, low-frequency, and peaks[11,32]—is not well reproduced by the quasiharmonic calculations. As explained in the previous paragraph, the likely cause is a failure of the harmonic mode assumption for excited states, rather than the DFT itself.

To understand the high-temperature phases, one needs to examine non-harmonic behaviour, including rotation, which means going beyond a single unit cell, e.g. using molecular dynamics. Molecular dynamics requires forces on each atom based on the positions of all the atoms in the system, which requires a force model which is fast enough to allow large simulations. Here we use a machine-learning approach to derive a transferable force model based on an interatomic potential. There are several approaches to machine-learning interatomic forces[33–35], which balance speed, transferability and accuracy. We adopt an approach focusing on transferability.

Any machine-learned potential should conserve energy, and therefore be based on a Hamiltonian (the potential). Forces are guaranteed to be conservative if they depend on translational and rotational invariant quantities: the "fingerprint" of each atom. We are interested in molecular phases here, so our potential specifies which atoms are "bonded" and allows stretching but not breaking of bonds.

In this paper we apply the machine-learning approach so create an interatomic potential for molecular hydrogen. We show that the potential describes the three molecular solid phases, with free-rotor Phase I, and broken-symmetry Phase II and a high-pressure Phase III. Furthermore, the melt line has a maximum, such that at high pressure the liquid is denser than the hcp solid, a feature we attribute to short-range directional order giving lower quadrupole–quadrupole interaction in the solid. The potential is trained on energies and classical Hellmann–Feynman forces derived from standard DFT in the Born–Oppenheimer approximation adopted by all standard DFT codes, so the interatomic potential is the same for deuterium, hydrogen-deuteride (HD) and hydrogen. The potential accounts for binding due to electronic structure: contributions from quantum-nuclear effects can be incorporated using lattice dynamics or path integral methods.

## Results

**Fitting forces and the phase diagram.** A particular challenge for hydrogen comes from the hierarchy of energies. The covalent bond is much stronger than the van der Waals attraction between molecules, which is turn is much stronger than the EQQ interactions which determine molecular orientations. To address this our potential combines a hierarchical fitting strategy alongside machine-learning (HMLP) described below.

For transferability testing, we used the standard approach of fitting to a subset of the data and testing against a different subset. Furthermore, we used an iterative fitting process: a trial potential was fitted, and applied in both Phase II annealing and melting line MD simulations. If novel configurations were found, they were used to generate more reference states for the DFT database, and the fitting process was repeated. This iterative process ensures that spurious structures are suppressed and the ground state structure is the same as found in DFT. Technical details of the forcefield parameterisation are given in the "Methods" section.

**Phase diagram.** Figure 1 shows the very good agreement between the classical HMLP and the DFT phase diagrams. By eliminating finite size effects, the HMLP can capture the full long-range correlations, however, this does not appear to have a significant effect on the phase boundaries.

In Phase I, the $H_2$ molecules exhibit free rotation at pressures below 40 GPa and temperatures below 900 K (orange hexagon symbols); at higher pressures the rotation is inhibited but there is no long-ranged orientational order. At low temperatures, Phase II becomes stable (red triangles), and the stable temperature region increases gradually with pressures. At high temperatures, the hcp lattice collapses to a liquid state. The calculated melting curve has a strong positive slope ($dT/dP > 0$) at low pressures, reaches a maximum at around 900 K and 90 GPa, and then drops. The HMLP predicted phase diagram agrees reasonably with experimental observations, as well as DFT (Fig. 1).

The HMLP and DFT predictions are good for the melt curve, but both overstabilise the broken-symmetry Phase II. This is due to the lack of quantum-nuclear effects, notably the zero-point energy, and can be addressed by including quantum-nuclear effects in the simulation. The discrepancy in the Phase I–II line does not indicate any inaccuracy of the HMLP itself. Our predicted melting curve is consistent with experiments: the value for the melting curve maximum is located between 80 and 100 GPa and 900 K, similar to the HMLP potential values. It also agrees with two-phase ab initio simulations, which proposed a gradual softening of the intermolecular repulsive interactions as its cause[16]. The close agreement of the HMLP transition pressure with experimental data enables us to accurately simulate behaviours of temperature- or pressure-driven phase transition between Phase I and II. A

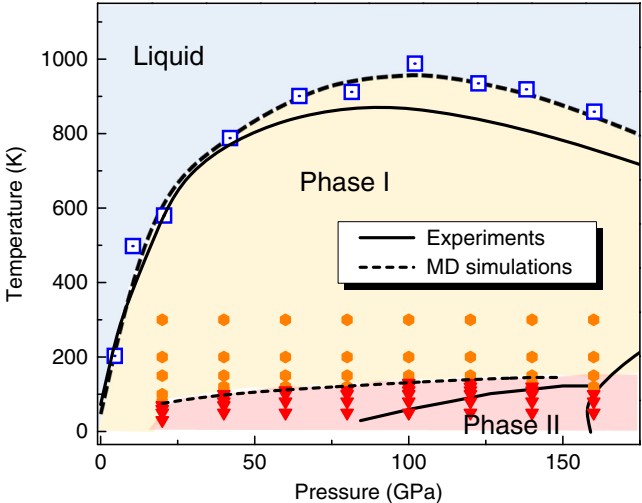

**Fig. 1 Machine learning interatomic potential simulated phase diagram of hydrogen.** Each datapoint represents a machine-learning interatomic potential based molecular dynamic simulation (HMLP-MD) which is itself in agreement with the equivalent DFT simulation. Blue squares represent the melting points form Z-method while the data of stable Phase I and Phase II appears as orange hexagon and red triangles, respectively. The dashed melt line and phase I–II boundary is to guide the eye. Experimental phase boundaries (solid lines) are taken from ref. [15].

"Phase III" is observed at higher pressures, corresponding to a different symmetry-breaking. However, by design the present HMLP model should start to fail to capture the properties of $H_2$ at still higher pressures, where molecule dissociation needs to be considered.

**Nature of Phase I**. Phase I can be easily recognised in MD by ordering of the molecular centres on the hcp lattice, and disorder of the orientations. Although frequently referred to as a free-rotor phase, we find this to be true only at low pressures. As pressure is increased the angular momentum autocorrelation becomes shorter than a single rotation, and then acquires a negative component, indicating that the molecule is librating.

Another characteristic of Phase I is the molecular vibration or "vibron": in Raman scattering this corresponds to the in-phase vibration of all molecules. The vibron frequency first increases, then decreases with pressure. Two plausible reasons are given for this reduction: either increased intermolecular coupling or weakening of the covalent bond. In our model, the covalent bond is always described by the same Morse parameters, so changes in the vibron frequency can arise only from resonant interactions between molecules, not weakening of the bond. Thus reproducing the reentrant vibron behaviour is a test of both the physical basis and the parameterisation of the model.

Since the molecules are rotating in Phase I, lattice dynamics cannot be used, so Raman phonon frequency is numerically characterised by the in-phase mode-projected velocity autocorrelation function (VAF)[36]. Trajectories and velocities were produced from 150 K HMLP-MD simulations within the micro-canonical ensemble (NVT) initiated in the $P2_1/c$-24 structure. A very fine time step of 0.05 fs was used and the trajectory and velocities were saved every ten time steps. By calculating the bond stretching velocities and Fourier transformation of the VAFs we calculate both the total vibron

density of states[26,37] from

$$g_{\text{tot}}(\omega) = \sum_{ik} \int [r_{ik}^2(t)] \exp i\omega t, \quad (1)$$

and the signal from the most strongly Raman-active mode,

$$g_{\text{Raman}}(\omega) = \int \left[ \sum_{ik} r_{ik}^2(t) \right] \exp i\omega t, \quad (2)$$

where $ik$ runs over all molecules (comprising atoms $i$ and $k$). A similar projection method is used for the E2g phonon[38].

Figure 2 plots the calculated total vibron spectra of solid and liquid hydrogen as a function of pressure from MD simulations. Both show a signature of vibron turnover above a critical pressure (about 54 GPa), consistent with the experimental observations[11]. This proves that bond weakening is not required for the turnover, since our potential has a fixed bond strength. Notably, the mean bond length in Phase I decreases monotonically with pressure (Fig. 2c), again at odds with ideas of bond weakening. What appears to be happening is a competition between two effects: at higher pressures the compression of the bond causes an increase in the frequency due to anharmonicity in the potential, whereas above 54 GPa the frequency is lowered due to coupling between the molecules.

The hcp structure has a Raman-active mode (E2g symmetry) corresponding to shearing motion of the basal plane. The frequency of this mode is experimentally well-determined and extremely pressure-dependent, from 36 cm$^{-1}$ at zero pressure to 1100 cm$^{-1}$ at 250 GPa[39–42]. Figure 2d shows a comparison of our potential with experimental pressure dependencies $\nu(P)$ of the E2g optical. The red symbols in are our HMLP-MD predictions, consistent with the DFT data of this mode extracted from our calculations. Comparing the calculations with experiment shows that the HMLP predicted frequency curves agree better with experiment than existing isotropic empirical potentials[43,44] (olive curve).

**Denser than close-packed liquid**. Figure 1 shows that the potential correctly reproduces the turnover and negative Clapeyron slope. We investigated the possible explanation for this denser than close-packed liquid. Figure 3a shows the equation of state for both solid and liquid phases, with the crossover indicating where the liquid is denser than the solid. The HMLP predicts a negative thermal expansion, which is consistent with DFT[45]. The normalised radial distribution function (Fig. 3b) shows that the liquid structure is essentially unchanged with pressure up to the pressures where bond breaking becomes a factor. We therefore deduce that the denser liquid is not related to the molecular–atomic transition.

Figure 4 compares the solid and liquid at the melting point. They are remarkably similar: close to the phase boundary the liquid shows five discernible neighbour peaks indicating short-ranged structure to 10 Å. The molecular bond length is longer in the liquid than the solid (shown more clearly in Fig. 2), but the separation between molecules is noticeably smaller in the liquid as evidenced by the first peak in the molecule–molecule RDF. This means that the molecules get closer together in the liquid, despite being longer.

Intermolecular interactions are dominated by quadrupole–quadrupole interactions and steric repulsion. Table 1 shows the implied contribution from quadrupole–quadrupole interactions calculated by electrostatics from HMLP simulations. Although the ML potential has no explicit electrostatic terms, there is a strong orientation correlation, which lowers the quadrupolar energy, not only in Phase II, but also in Phase I and to a lesser extent in the liquid.

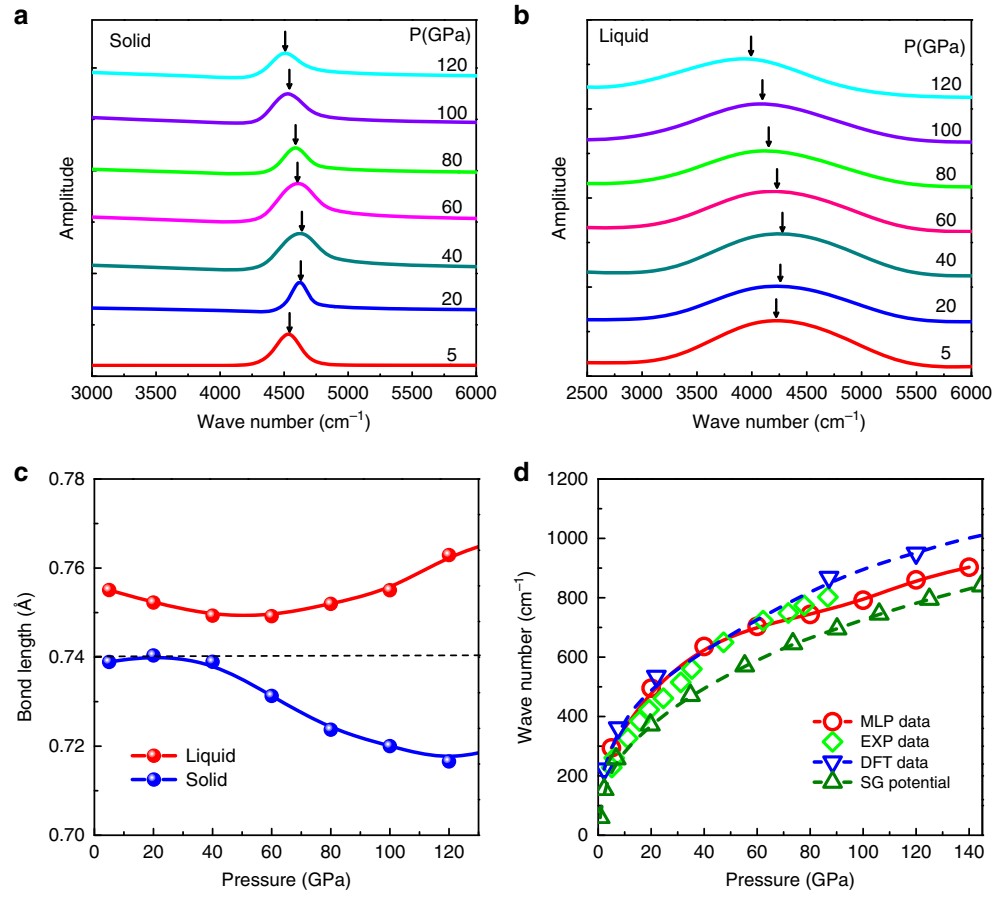

**Fig. 2 Calculated Raman signals of solid and liquid $H_2$ as a function of pressures. a** Pressure dependence of vibron frequencies from the solid $H_2$ at $T = 150$ K, arrows emphasise the turnover of the peak frequency. **b** Pressure dependence of vibron frequencies from the liquid $H_2$ at $T = 1000$ K. **c** The mean H–H bond length as a function of pressures, the dotted horizontal line is to guide the eye. **d** The pressure-dependent E2g phonon frequency compared with our DFT, recent experiments[58,46], and Silvera–Goldman potential[43,44].

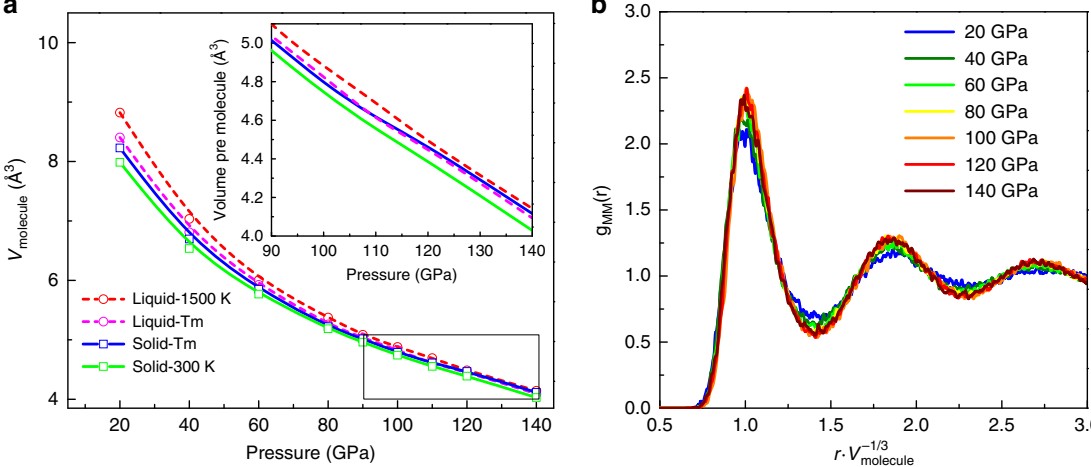

**Fig. 3 Structural properties and EOS of liquid $H_2$ at different densities. a** Equation of state (EOS) for liquid and solid $H_2$ at selected temperatures. The green and blue solid line is the EOS of Phase I at 300 K and $T_m$ (melting point), respectively. The pink and red dash line is the EOS of liquid at $T_m$ and 1500 K, respectively. **b** Normalised radial distribution function of molecular centres at selected pressures and $T = 1000$ K, indicating no liquid–liquid phase transition below 140 GPa.

We hypothesised that molecules can get closer together if their constituent atoms are further apart (i.e. in an "X" shape viewed down the intermolecular vector). By contrast, the solid has orientations which offer higher cohesive energy.

Figure 5b investigates this further using DFT, showing that the T configuration which optimises the quadrupole interaction becomes unfavourable with respect to the X configuration at a separation of 2.25 Å. As we saw in Fig. 4, the nearest neighbours are already this close by 20 GPa.

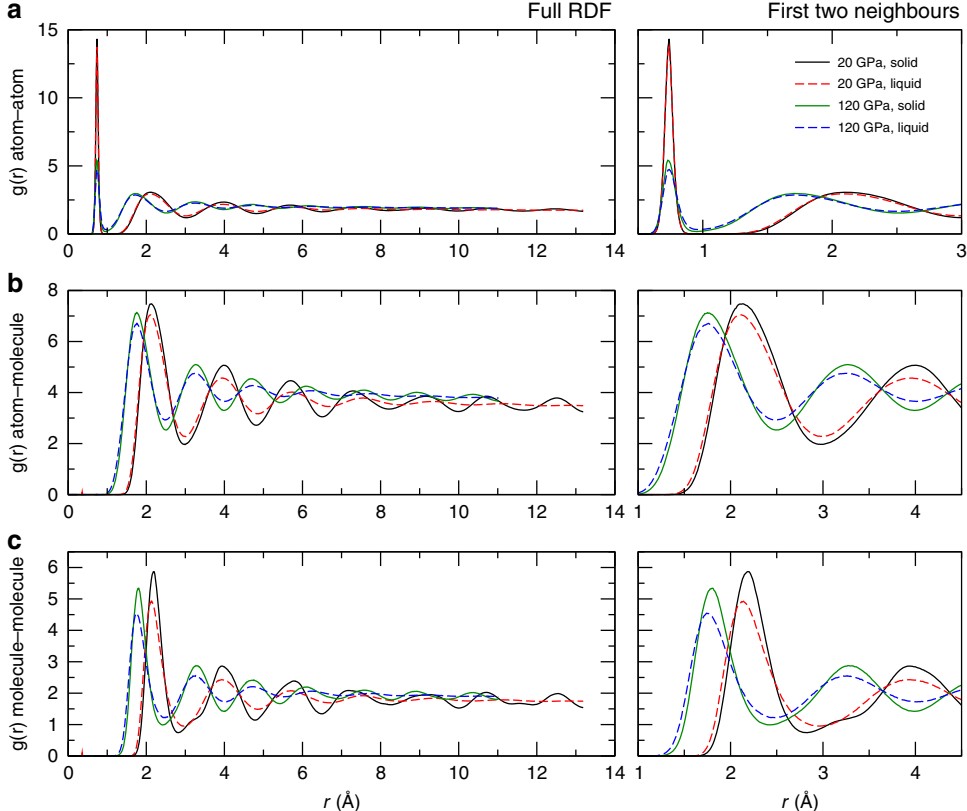

**Fig. 4 Radial distribution functions (RDFs) of co-existing solid and liquid. a** Atom–atom RDFs, **b** atom–molecule RDFs and **c** molecule–molecule RDFs, with the full length shown on the left and a close-up of the first two neighbour shells on the right. The RDFs show that the liquid can pack more tightly than the hcp solid phase. This is true both at 20 GPa, which is before the turnover in the Clapeyron slope, and at 120 GPa, well into the negative slope region. The higher density of the liquid is most apparent in the molecule–molecule RDF. These RDFs are on good agreement with the RDFs obtained from AIMD simulations[45].

| Table 1 EQQ for liquid and solid phases. | | | | |
|---|---|---|---|---|
| Pressure (GPa) | $<E_{QQ}^{L}>_{melt}$ | $<E_{QQ}^{S}>_{melt}$ | $<E_{QQ}^{S}>_{PhaseI}$ | $<E_{QQ}^{S}>_{PhaseII}$ |
| 20 | −5.5897 | −6.2366 | −15.5999 | – |
| 40 | −7.3219 | −7.7636 | −25.4964 | −43.4700 |
| 60 | −8.9744 | −9.8332 | −35.9825 | −52.9075 |
| 80 | −10.9062 | −13.6786 | −44.4900 | −65.9290 |
| 100 | −11.7181 | −15.9932 | −53.0947 | −76.0736 |
| 120 | −11.8199 | −17.7660 | −59.5902 | −91.3044 |
| 140 | −11.8821 | −18.5865 | −65.5554 | −103.4814 |

The EQQ for liquid and solid phases at the melt points, alongside values calculated for stable Phase I ($T = 150$ K) and Phase II ($T = 50$ K) structures. All values are in units of meV/molecule. For uncorrelated free rotors, $<E_{QQ}> = 0$

To quantify this, we looked across the various fingerprints to see which most strongly differentiate liquid from solid configurations on the melt line. This turns out to be $fp\_E_{QQ}$, the fingerprint with the same functional form as the quadrupole–quadrupole interaction: the MLP has learned that this is important. Figure 5a shows that the contribution from $fp\_E_{QQ}$ becomes significantly higher in the solid above 70 GPa. We note that $fp\_E_{QQ}$ is dimensionless, and its fitted contribution to the potential is different from the estimated quadrupole–quadrupole energy compared in Table 1.

These findings explain why the liquid is denser than the solid. At pressures above the melting point maximum, orientation-dependent interactions are strong enough that rotation is inhibited[46–48]. However, these low-energy arrangements are typically associated with larger intermolecular distances (e.g. the

T configuration). By contrast, the liquid favours other arrangements which allow the molecules to come closer.

The X configuration maximises the atom–atom distance for a given intermolecular separation. This and similar arrangements compensates for the smaller molecule–molecule distance in the liquid to give the same peak position in the atom–atom RDF in both liquid and solid.

**Nature of Phase II**. We performed extensive HMLP-MD simulations around the Phase I–II boundary, with different starting configurations, to determine candidate structures and phase stability of $H_2$ Phase II. At 150 K and 20 GPa, the orientations of the $H_2$ molecular axes are almost randomly distributed along different directions, indicating Phase I with freely rotating molecules. Upon compression to the high pressure of 80 GPa and cooling to a low temperature of 50 K, the material transforms to an orientationally ordered phase in which the molecular rotations are restricted (Fig. 1). By carefully comparing it with candidate structures proposed by ab initio calculations, we find that the lattice and molecular ordering is close to $P2_1/c$-24, which has been one of the most thoroughly studied and strongest candidate for Phase II[5,22,23,49].

The HMLP does not include quadrupole interactions explicitly —it has "learnt" them. Table 1 shows what the quadrupole interactions would be, using electrostatic calculation based on the HMLP configurations. The large negative values indicate the prevalence of quadrupole-type ordering: strongest in Phase II. The differences, tens of meV, is of similar magnitude to the phase

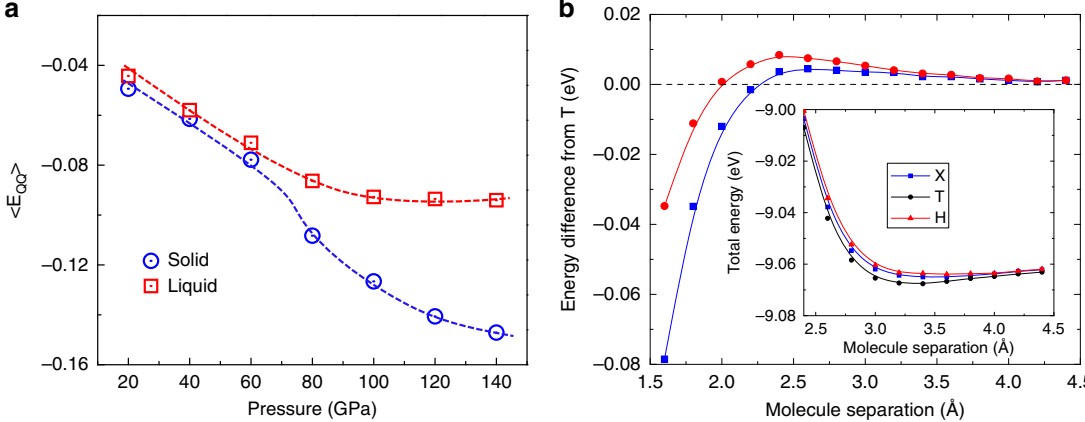

**Fig. 5 Local hydrogen molecule arrangement of solid and liquid phases. a** The contribution of the $fp\_E_{QQ}$ fingerprint in solid and liquid. Data were obtained as a time average of single-phase calculations the solid–liquid phase boundary. **b** DFT calculations of the interaction between two hydrogen molecules as a function of distance for three orientations. In the DFT calculations the separation vector lies along the z-axis, and one of the molecules to point along x; H, X, and T represent the second molecule pointing along x, y, and z, respectively. The configuration with both pointing along z is always unfavourable. Main figure: energy differences relative to $T$, Inset: total energy relative to free atoms.

transformation temperature. The HMLP has learnt that Phase II is stabilised by quadrupole interactions.

The corresponding RDF indicates that the centre of each molecule remains close to the hcp lattice sites. Consequently, we define an order parameter $O$ relating the structures of Phase II in our HMLP-MD simulations with the static-lattice DFT predictions of $P2_1/c$-24. The average value $\langle O \rangle$ exhibits a sharp change as the system transitions from the structured Phase II to the rotationally symmetric Phase I with increasing temperature. Transition temperatures were taken at discontinuous jumps in $\langle O \rangle$. This produces the phase diagram shown in Fig. 6b. Note that the transition temperatures obtained from analysis of $\langle O \rangle$ agree with those obtained from peaks in the heat capacity calculated as $\left(\frac{\partial H}{\partial T}\right)_P$ from finite differences (see Supplementary Fig. 5). The phase boundary agrees well with the experiment[32], particularly for the more classically behaved deuterium. Similarity to experiments on deuterium rather than hydrogen is perhaps unsurprising, since nuclear quantum effects such as zero-point motion are significant at the low temperatures investigated here.

**Transition to Phase III**. Above 160 GPa we find a high-pressure transformation to a broken-symmetry structure different from Phase II dominated by efficient packing rather than EQQ, this is at approximately the same pressure as Phase III.

Experimentally, Phase III is associated with a sharp drop in the vibron frequency and the appearance of a strong IR signal. This implies a non-centrosymmetric structure and a weakening of the molecules. In studies of HD a process of bond dissociation and recombination ("DISREC", $2HD \rightarrow H_2 + D_2$) has been observed[50]. This bond breaking is seen in DFT to also occur in pure $H_2$[26,30]. Our potential does not allow for bond breaking, so we have not studied the dynamics of this "Phase III" in detail.

## Discussion

In summary, we have introduced a heirarchical, iterative machine-learning based interatomic potential for atomistic simulations of $H_2$ molecules, by directly learning from reference ab initio molecular dynamics simulations. The resultant HMLP-MD approach predicts angular energy dependence in the range of tens of meV/atom and demonstrates good transferability to various structural environments. Several applications have been presented for which our potential is particularly well suited. The fast, transferrable potential is also suitable for a wide range of

further applications and extensions, including compounds, bond breaking, and path integral calculations.

The simulations reproduce the equilibrium temperature–pressure phase diagram for molecular phases (I, II, III, and melt, $P < 160$ GPa). The maximum in the melt curve in hydrogen is highly counterintuitive—it requires that the liquid is denser than the hexagonal close-packed solid. By detailed simulation we resolve this by showing that certain molecular orientations (e.g. X) allow the molecules to approach more closely, while others (e.g. T) have lower quadrupole energy. By monitoring an order parameter corresponding to the quadrupolar interactions find that, at high pressures, Phase I develops intermolecular correlations which lower the energy, while the liquid has correlation which lower the volume.

The simulations of the Phase I–II boundary show a transition from an ordered Phase II to a orientation-disordered Phase I. The molecules in Phase I are not freely rotation, and we find that rotation about the $c$-axis persists to higher-pressure/lower-temperature than rotation out of plane.

Our HMLP potential also has shown the capability of predicting the pressure dependence of the Raman-active E2g mode, consistent with experiment and previous DFT calculations. We explain the maximum frequency of the vibron as due to competition between molecular compression and stronger intermolecular coupling. Weakening of the covalent bond is not required.

## Methods

### Machine-learned interatomic potential

*Learning dataset*. Structures for reference atomic environments and benchmarks were accumulated from DFT-based ab initio MD runs. The DFT calculations were performed using the CASTEP package[51] within the Perdew–Burke–Ernzerhof generalised gradient approximation (PBE)[52] for the exchange–correlation function. A cutoff energy of 1000 eV for the plane-wave basis set and a k-point mesh of $1 \times 1 \times 1$ were selected. To ensure the transferability of the potential to a wide variety of atomistic situations, $H_2$ in different geometric arrangements was considered, including modest-sized bulk samples in Phase I, II, and liquid, composed of 144 $H_2$ molecules. Moreover, unusual configurations found in HMLP-MD with preliminary versions of the potential were added to the DFT training set to improve performance and transferability. The final accumulated dataset includes up to 41,468 configurations, which are provided as a separated file in the Supplementary Information.

The PBE functional was chosen because it has become the de facto standard in studies of molecular systems, and high-pressure hydrogen. PBE has been criticised for overstabilising the metallic phases: this is due to its behaviour at high electron density gradients[53], and not vdW corrections. In this work we only consider molecular phases, so this is not a concern.

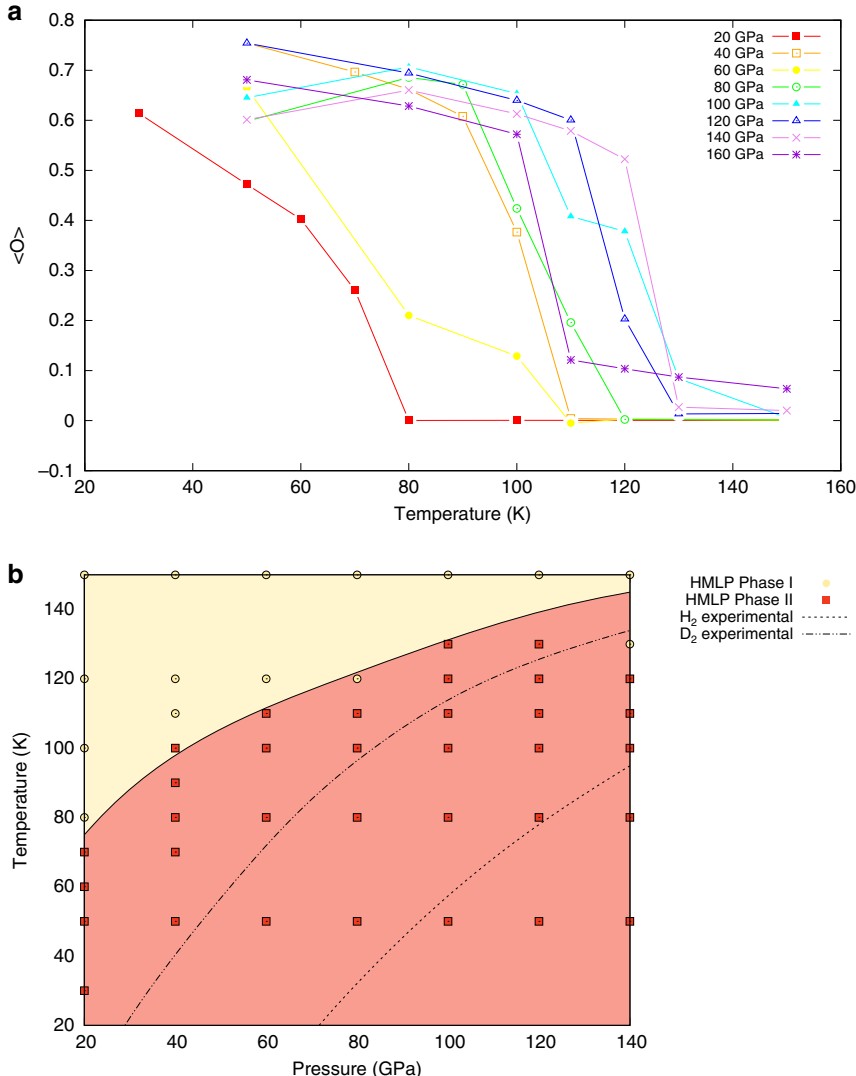

**Fig. 6 Order parameter and phase boundary for the I–II transition. a** Order parameter $\langle O \rangle$ as a function of temperature for the seven pressures investigated in this work. In all cases there is a sharp decrease from an ordered system ($\langle O \rangle = 1$) to a disordered system ($\langle O \rangle = 0$). The system is considered to be Phase I after $\langle O \rangle$ drops discontinuously below 0.1. **b** The resultant phase boundary for the I–II transition in the classical solid. The dashed lines represent the experimental phase boundaries for hydrogen and deuterium, respectively[32].

We emphasise that determining the suitable dataset is not straightforward. Numerous iterations of the potential were required to obtain a good fit to the phase diagram. A good fit to DFT energies of known phases is not evidence that other phases are unstable: we test each iteration of the potential by running MD simulations in NPT ensemble, cycling the pressure and temperature between the regions expected for Phases I, II, and III to ensure that all phase transitions were between the fitted phases. When a crystal structure different from the fitted one was found, this new phase was calculated using DFT. If this showed that the ML potential described the phase poorly, it was added to the training set and the training redone. The success of such strategy is supported by an example shown in Supplementary Note 2.

*Covalent bond.* The covalent bonding contribution to the force is approximated as $(F_1^{u_{12}} - F_2^{u_{12}})/2$, where $F_1^{u_{12}}$ is the component of atomic force projected down the molecular axis. We examined various options for fitting the covalent bond: harmonic, Lennard-Jones and Morse potentials. Although the Morse form provides the best fit across our dataset using simple regression, we adopt the harmonic form in our potential to prevent artificial bond-breaking events at high pressure (>120 GPa). Supplementary Table 1 shows the fitted parameters for the harmonic and Morse form, respectively.

*Non-bonded interactions.* We describe the short-ranged Coulomb and van der Waals potentials using pairwise functions to create the fingerprint. These are built using Gaussians with a smooth cutoff in the form

$$V_i^k = \sum \exp(-|r_{ij}/\eta_k|^2) f_{cut}(r_{ij}), \qquad (3)$$

combined with damped sinusoidal function form

$$V_i^k = \sum \sin(k r_{ij}) \exp(-r_{ij}/\eta_k) f_{cut}(r_{ij}), \qquad (4)$$

where $\mathbf{r}_{ij}$ is the distance between atom $i$ and $j$ with $\eta_k$ the range of the $k$th fingerprint, and $f_{cut}(r_{ij})$ is a damping function for atoms within the cutoff distance.

These fingerprints are mapped onto the corresponding residual energies, defined by the DFT energies less the contribution from the covalent bonding contribution.

This mapping is achieved using the kernel ridge regression (KRR) method which is capable of handling complex nonlinear relationships[54]. The details of parameterisation and mapping algorithms are shown in Supplementary Note 3.

*Orientation-dependent interactions.* The orientation-dependent contribution is the last considered. These are fitted to the residuals once covalent and pairwise interactions are subtracted from the DFT energies. The corresponding fingerprints for the orientation-dependent interactions are listed in Supplementary Note 3 and Supplementary Table 2.

The fidelity of our HMLP is evaluated by the comparison of our ML prediction and the DFT calculations in Supplementary Fig. 6. The mean absolute error is of the order of the expected numerical and theoretical accuracy of the reference quantum mechanics-based calculations, indicating good performance of the present ML model.

**Machine-learned molecular dynamics**. The simulations were performed using periodic boundary conditions and a time step of 0.5 fs. The Nose–Hoover thermostat and the Parrinello–Rahman barostat[55] were used for controlling temperature and pressure, respectively. All simulations were carried out using the LAMMPS package and the atomic configurations were visualised with the AtomEye program. Typical models of the $H_2$ system was created with $P2_1/c$-24 structure containing up to 72,576 molecules. To reproduce the entire temperature–pressure phase diagram, the NPT simulations of 1152-atom supercells of $P2_1/c$-24 structure were carried out at selected temperatures and pressures, from which we can identify the corresponding stable phases and melting point via the Z-method[56]. Furthermore, the phase-coexistence method[57] with co-existing 27,648 molecules of $H_2$ solid and liquid was adopted to determine the properties of solid and liquid phases at the melting curve. To probe the Phase I–II boundary, 2304 atom supercells of the Phase II $P2_1/c$-24 structure were allowed to equilibrate for 250 ps at a series of pressures and temperatures.

**Analysis of molecular dynamics**. To distinguish between the broken-symmetry structure of Phase II and the free rotors of Phase I we introduce the orientational order parameter $O$.

$$O = \left\langle \frac{\sum_b \sum_{i \neq j} (\hat{\mathbf{r}}_{i,b} \cdot \hat{\mathbf{r}}_{j,b}) \mathbf{R}_{ij,b}}{\sum_b \sum_{i \neq j} \mathbf{R}_{ij,b}} \right\rangle. \tag{5}$$

Here the summation is over unit cells $i, j$, each containing a set of basis molecules $b$. Unit vectors $\hat{\mathbf{r}}_{i,b}$ and $\hat{\mathbf{r}}_{j,b}$ are oriented along the the H–H bond of the $b$th molecule in the $i$th and $j$th unit cell, respectively, and $\mathbf{R}_{ij,b}$ is the distance between the centre of mass of these two molecules. The angled brackets denote a time average. This parameter probes the long-range order in the system relative to the chosen basis, which in our case is the $P2_1/c$-24 unit cell[23]. A value of 1 means that the system has the $P2_1/c$-24 structure, and a value of 0 suggests that the system is disordered. Note that this order parameter only detects similarity to the given basis and thus a phase change to a structure with a different unit cell will yield an erroneously low value. The trajectories were therefore visually inspected in addition to the order parameter analysis. A $4 \times 6 \times 4$ supercell of $P2_1/c$-24 (2304 atoms) was used for Phase II, and the unit cells and basis for this system are illustrated in Supplementary Fig. 7. MD trajectories were calculated for temperatures ranging from 30 to 150 K and pressures from 20 to 160 GPa. After a 50 ps equilibration period, the order parameter $O$ was averaged over the remaining 200 ps of the trajectory. The results are shown in Fig. 6a.

The Raman-active phonons were extracted from the MD using the projection method which automatically includes anharmonic effects[36,38].

We tried numerous approaches to measure the orientation relationship between adjacent molecules, $i$ and $j$ with interatomic vectors $\vec{\sigma}_i$ and $\vec{\sigma}_j$ separated by $\vec{R}$. For Table 1 we used the explicit equation for linear quadrupoles:

$$E_{QQ} = \frac{3Q^2}{4\pi\epsilon_0} \sum_{i,j} \frac{\Gamma(\vec{\sigma}_i, \vec{\sigma}_j, \hat{R})}{|\vec{R}|^5}, \tag{6}$$

where $Q = 0.26$ DÅ is the quadrupole moment of the $H_2$ molecule and the orientational factor $\Gamma(\vec{\sigma}_i, \vec{\sigma}_j, \hat{R})$ is defined as:

$$\begin{aligned} \Gamma(\vec{\sigma}_i, \vec{\sigma}_j, \hat{R}) = {} & 35(\vec{\sigma}_i \cdot \hat{R})^2 (\vec{\sigma}_j \cdot \hat{R})^2 - 5(\vec{\sigma}_i \cdot \hat{R})^2 - 5(\vec{\sigma}_j \cdot \hat{R})^2 \\ & + 2(\vec{\sigma}_i \cdot \vec{\sigma}_j)^2 - 20(\vec{\sigma}_i \cdot \hat{R})(\vec{\sigma}_j \cdot \hat{R})(\vec{\sigma}_i \cdot \vec{\sigma}_j) + 1. \end{aligned} \tag{7}$$

## Data availability

Datasets used in generation of the $H_2$ potential are available at the Edinburgh DataShare https://doi.org/10.7488/ds/2874.

## Code availability

Plugin code with which the interatomic potential can be implemented through LAMMPS is available as supplementary online material in Supplementary Dataset 1.

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

## Acknowledgements
The authors acknowledge the ERC project HECATE for funding. We are grateful for computational support from the UK national high performance computing service, ARCHER, and from the UK Materials and Molecular Modelling Hub, which is partially funded by EPSRC (EP/P020194), for both of which access was obtained via the UKCP consortium and funded by EPSRC grant ref. EP/P022561/1. We thank Miriam Pena-Alvarez and Eugene Gregoryanz for the experimental phonon and vibron data. We thank Hua Geng for sharing complete DFT data from ref. [45].

## Author contributions
G.J.A. conceived the study, H.Z. coded and performed the machine-learning procedure, H.Z. and G.J.A. generated ab initio data, H.Z. and H.W. performed molecular dynamic simulations, all authors analysed the simulation results, and contributed equally to writing the manuscript.

## Competing interests
The authors declare that they have no competing interests.
