## [Peer Review File · Nature Communications]

REVIEWER COMMENTS

Reviewer #1 (Remarks to the Author):

The manuscript presents a new machine-learning based classical potential suitable for simulation of molecular phases of hydrogen at high pressure. Employing this potential for MD simulations, several findings concerning the solid and liquid region of the high-pressure phase diagram of molecular hydrogen are presented. The most important one appears to be the explanation of the turnover of the Clapeyron slope around 100 GPa. The authors propose that this is related to the change of the mutual orientation of neighboring molecules which is calculated from MD simulations of solid and liquid along the coexistence line. The quantity used to measure this orientation in solid and liquid indeed exhibits a crossing around 80 GPa and this is presented as explanation for liquid being denser than solid at high pressures. To me, this explanation is not convincing enough. The difference in the X measure presented in Fig.5a is extremely small, reaching value of 10^{-3} at the lowest pressure of 20 and highest pressure of 140 GPa. This value is in my opinion unlikely to fully explain the difference in density between solid and liquid which according to Fig.3a appears to be of the order of 1%. I am afraid that the X measure is not a relevant quantity in this respect and the explanation of the origin of the change of the Clapeyron slope might require also other considerations.

Another point that does not convince me is the determination of the I-II transition temperature presented in Fig.6. The authors employ the simple procedure based on monitoring the orientational order parameter upon heating the system. According to the information presented in Methods a 5 ps equilibration period was applied. This time is more typical for ab initio simulations and with the classical potential far longer equilibration times could be used. Since the transition is of first order it is likely that overheating is present and therefore the approximate agreement with experimental phase boundary for deuterium shown in phase diagram in Fig.6b might be accidental.

The text contains some long sentences which are hard to understand, e.g.

p.9

'For a given intermolecular separation, the X configuration maximizes the atom-atom distances this and similar arrangements compensates for the smaller molecule-molecule distance in the liquid to give the same peak position in the atom-atom RDF in both liquid and solid.'

or caption of Supplementary Figure 3

'Phase lines are drawn based on a visualization of the trajectory as discontinuous jumps in O for the higher pressure cases are due to phase changes within phase II and not a change to phase I.'

In Methods, the name of the last author of the PBE functional is Ernzerhof.

I am not convinced that the findings presented in the manuscript justify its publication in Nature Communications.

Reviewer #2 (Remarks to the Author):

This paper is a fine demonstration of how machine learned (ML) force fields can be used to study complicated condensed phase systems, and the topic of application, high-pressure hydrogen, is one such system of wide interest in different areas of physics. The authors develop a ML force field that is used to run calculations that would have previously required density functional theory or even more accurate methods, then using this force field to calculate structural properties of hydrogen at various state points. That said, I believe that the work currently falls short of what the title suggests, namely "Understanding high pressure hydrogen" in its entirety. I believe that more detailed insight could be gained with relative ease from the authors' method and data, and more details need to be given for others to reproduce the work.

I feel that the methodology itself (using some variants of ML force fields) is to some extent routine by now, but the application to an interesting system could be expected to be of interest to a wide audience. I would therefore recommend to consider further a suitably (and substantially) extended manuscript. I am aware that the current situation is imposing constraints on all of us, so given below are recommendations primarily with regard to the data analysis and presentation, rather than expecting the authors to run extensive new calculations.

(1) The idea of a "denser than close-packed" liquid (l. 17 and main text) is interesting, although in some sense misleading. Close packing refers to atomic spheres, and no denser packing than close packing can be achieved. In the case the authors are referring to, the structure is built up from molecules (with more empty space) and the fact that a liquid can be denser than this does not appear to be so surprising to me. In turn, the authors' method presumably allows them to gain deeper insight into the local structure of this liquid than what is commonly done (the manuscript currently focuses on rather simple radial distribution function plots). Is there another, possibly more

creative way, of obtaining a deeper knowledge of what this liquid phase looks like (e.g., other order parameters and ways to quantify the orientation)? What have others in the field done?

(2) It is mentioned briefly that "Our potential does not allow for bond breaking" (p. 259), which appears to be strange to me. If the ML force field is able to reproduce a DFT energy landscape, then it should be able to break bonds as well (or badly) as DFT describes it. This would be relevant for fully describing the phase diagram, and one would hope to read at least a thorough justification why the bond breaking could not be achieved despite the authors' best efforts and why this does not affect any of the other results.

(3) Minor aspects of presentation: It seems strange to state that "Most information is gleaned from spectroscopy" (l. 34) and then to cite two papers which are strongly reliant on diffraction rather than spectroscopy. In Fig. 3a, it is not explained what the dashed line represents and the labels of the inset are very difficult to read. Aspects of the method such as "the projection method" for Raman predictions (l. 359) should be explained in more detail, so that readers can better understand what is being done here.

(4) Data/Methods availability: I do not find the data availability statement that "All relevant data are available from the authors" (l. 388) to be fully satisfactory. In fact, it is not easily possible to understand from the Methods section how exactly the potential was fitted, much less to completely reproduce the authors' work. This is important especially given the apparent combination of classical and ML fitting methods (which is interesting!). The very terse presentation in the Methods section also leads to open questions on the side of the reader, e.g. when "the short-ranged Coulomb and van der Waals potentials" (l. 311) are mentioned - surely both these types of interaction are long-ranged? In summary, full details of the method should be given (those parts that do not fit within the length limit of the Methods could be placed in the SI), and it should be clarified whether the LAMMPS implementation will be published, and when, because this would strongly increase the usefulness of the authors' work to other researchers.

Reviewer #3 (Remarks to the Author):

The manuscript reports the results of MD simulations of molecular hydrogen with a potential fitted to DFT energies. They explain many of the unusual properties of hydrogen at high pressures. I think the research is a valuable contribution and should be published after the authors provide more details about their calculation.

The potential was constructed using the PBE density functional. Can the authors justify the use of this functional? Note PBE does not include the van der Waals interaction and has been shown to be less accurate for hydrogen than those functionals which do have VDW interactions. Perhaps line 99 "long-range interactions" should be clarified.

I am concerned that the study is not reproducible. Neither the detailed procedures of fitting the DFT potentials nor the results of fitting are available. I do not regard communication with the authors as suitable alternative.

-In doing the fit what is the objective function? energies or forces or both?

-what are the calculated values of the Morse potential?

-what are the parameters in Eq 3 for the non-bonded potential?

-Provide details of the Kernel Ridge regression?

-How many configurations are in the training set?

On line 305 It is mentioned that "rigorous testing in MD is essential" Can the authors be more precise?

-How good is the fit to DFT forces? (e.g. the rms value of the error in atomic units) Using this value another author could compare their ML potential.

Responses to Reviewers' Reports on NCOMMS-20-14026

We thank all the three reviewers for their careful review of this manuscript. We are pleased that both Reviewers 2 and 3 are largely supportive of this paper for publication in Nature Communications, aside from reasonable requests regarding the code/data availability and other a few minor questions or issues to be further clarified.

Reviewer 1 also only raised a few minor places to be further improved, but has two main concerns before our paper can be published. First, Reviewer 1 is concerned that the difference in the X measure is not large, unlikely to fully explain the difference in density between solid and liquid. Secondly, the 5 ps equilibration times of the MD simulation is too short to determine the I-II transition temperature.

As detailed below, we have addressed all the minor questions or issues raised by the three reviewers, and have revised the paper accordingly. As for the code/data availability, we have provided all DFT trajectory data online at our University site. The complete details of the machine learning model and the code for LAMMPS implementation are included in this resubmission.

With regard to the X measure issue, we realised that the orientational fingerprints of the MLP provide a better way to distinguish configurations of liquid and solid. Among all the orientational fingerprints, the one which has the form of the quadrupole-quadrupole energy ($fp-E_{QQ}$), has much stronger signal in solid than liquid, thus providing a better quantitative measure than the X measure.

In response to the I-II transition temperature issue, we repeated the calculations with larger sample size (up to 1152 atoms) and longer simulation times (up to 250 ps).

With these responses and improvements, we hope the paper is now acceptable for publication in Nature Communications.

Detailed responses to Reviewer 1

Comment 1: *The manuscript presents a new machine-learning based classical potential suitable for simulation of molecular phases of hydrogen at high pressure. Employing this potential for MD simulations, several findings concerning the solid and liquid region of the high-pressure phase diagram of molecular hydrogen are presented. The most important one appears to be the explanation of the turnover of the*

Clapeyron slope around 100 GPa. The authors propose that this is related to the change of the mutual orientation of neighboring molecules which is calculated from MD simulations of solid and liquid along the coexistence line. The quantity used to measure this orientation in solid and liquid indeed exhibits a crossing around 80 GPa and this is presented as explanation for liquid being denser than solid at high pressures. To me, this explanation is not convincing enough. The difference in the X measure presented in Fig.5a is extremely small, reaching value of 10^{-3} at the lowest pressure of 20 and highest pressure of 140 GPa. This value is in my opinion unlikely to fully explain the difference in density between solid and liquid which according to Fig.3a appears to be of the order of 1%. I am afraid that the X measure is not a relevant quantity in this respect and the explanation of the origin of the change of the Clapeyron slope might require also other considerations.

Figure R1 | The contribution of the fp_E_{QQ} fingerprint in solid and liquid. Data was obtained as a time average of single-phase calculations the solid-liquid phase boundary.

Response: We recognise that the difference in the X measure is not large, and have looked for a different measure of the same effect. We realised that the orientational fingerprints of the MLP provide a way to distinguish configurations. We have looked to see which fingerprints give the biggest difference between solid and liquid. Three stand out, the fp_E_{QQ} which has the form of the quadrupole-quadrupole energy; $fp1$ the angle between the projection of two molecules down the intermolecular vector; $fp2$ which is the angle between molecule and intermolecular vector. In each case the liquid has a higher value for the configuration maximised by X or minimised by T orientation. The fp_E_{QQ} - minimised by T - is largest and has much stronger signal in solid than liquid.

Thus, our original idea seems correct, and the fp_E_{QQ} provides a better quantitative measure than the X measure. Better still, the difference only becomes prominent at pressures where the liquid is

unexpectedly dense. We rewrote this section and thank the referee for pressing us to find this far more convincing presentation than our previous "X-measure".

Comment 2: *Another point that does not convince me is the determination of the I-II transition temperature presented in Fig.6. The authors employ the simple procedure based on monitoring the orientational order parameter upon heating the system. According to the information presented in Methods a 5 ps equilibration period was applied. This time is more typical for ab initio simulations and with the classical potential far longer equilibration times could be used. Since the transition is of first order it is likely that overheating is present and therefore the approximate agreement with experimental phase boundary for deuterium shown in phase diagram in Fig.6b might be accidental.*

Response: We appreciate the reviewer's desire to be further convinced about Phase I-II and the melt line. As the reviewer's suggestion, we repeated the I-II calculations with larger sample size and longer simulation times (2304 atoms, 250 ps with 50 ps discarded as equilibration). Then, to make doubly sure, we applied the Z-method to recalculate melting using single-phase calculations: the results are consistent with 27648-atom phase-coexistence calculations.

Comment 3: *The text contains some long sentences which are hard to understand, e.g. p.9 'For a given intermolecular separation, the X configuration maximizes the atom-atom distances this and similar arrangements compensates for the smaller molecule-molecule distance in the liquid to give the same peak position in the atom-atom RDF in both liquid and solid.' or caption of Supplementary Figure 3 'Phase lines are drawn based on a visualization of the trajectory as discontinuous jumps in O for the higher pressure cases are due to phase changes within phase II and not a change to phase I.'*

Response: We simplified these sentences

Comment 4: *In Methods, the name of the last author of the PBE functional is Ernzerhof.*

Response: We thank the reviewer for correcting our inaccurate statement. We have fixed it in the revised manuscript.

Detailed responses to Reviewer 2

General Comments: *This paper is a fine demonstration of how machine learned (ML) force fields can be used to study complicated condensed phase systems, and the topic of application, high-pressure hydrogen,*

is one such system of wide interest in different areas of physics. The authors develop a ML force field that is used to run calculations that would have previously required density functional theory or even more accurate methods, then using this force field to calculate structural properties of hydrogen at various state points. That said, I believe that the work currently falls short of what the title suggests, namely "Understanding high pressure hydrogen" in its entirety. I believe that more detailed insight could be gained with relative ease from the authors' method and data, and more details need to be given for others to reproduce the work. I feel that the methodology itself (using some variants of ML force fields) is to some extent routine by now, but the application to an interesting system could be expected to be of interest to a wide audience. I would therefore recommend to consider further a suitably (and substantially) extended manuscript. I am aware that the current situation is imposing constraints on all of us, so given below are recommendations primarily with regard to the data analysis and presentation, rather than expecting the authors to run extensive new calculations.

Response: We are pleased that Reviewer 2 confirms the novel and interesting aspects of this work. Below we respond to the constructive suggestions and comments on the presentation of this work, including the reproducibility issue. In this resubmission, we have done substantially extended calculations and additional analysis, provided all DFT trajectory data, the complete machine learning model and the HMLP code for LAMMPS implementation.

Comment 1: *The idea of a "denser than close-packed" liquid (l. 17 and main text) is interesting, although in some sense misleading. Close packing refers to atomic spheres, and no denser packing than close packing can be achieved. In the case the authors are referring to, the structure is built up from molecules (with more empty space) and the fact that a liquid can be denser than this does not appear to be so surprising to me. In turn, the authors' method presumably allows them to gain deeper insight into the local structure of this liquid than what is commonly done (the manuscript currently focuses on rather simple radial distribution function plots). Is there another, possibly more creative way, of obtaining a deeper knowledge of what this liquid phase looks like (e.g., other order parameters and ways to quantify the orientation)? What have others in the field done?*

Response: We thought further the reviewer's suggestion to make a more detailed investigation with "other order parameters". In a previous thesis by Ioan Magdau, many different 3-site measures were developed to distinguish crystal structures (see <https://era.ed.ac.uk/handle/1842/22047?show=full>), but

these typically exploit the crystal symmetry and well-defined neighbours and we didn't find them useful. To describe relative molecular orientations in the liquid, such order parameters must be four-body. This was done using the "X-measure" in the original manuscript, but it is possible to go further.

As described in reply to Ref 1, the fingerprints of the ML potential define a set of rotationally-invariant order parameters with appropriate symmetry. We compared weights of each fingerprint in the liquid and solid phase, and found that the biggest difference is in the fingerprint with the symmetry of the quadrupole-quadrupole interaction. This is similar to our previous idea that the solid favoured the "T" and the liquid favoured the "X" configurations, but more systematic in that we have rejected numerous other possible order parameters.

We don't think calling Phase I a "close-packed solid" is misleading. Although hydrogen is a diatomic molecule, phase I is always regarded as comprising free rotors, which are spherical in either classical time-averages or quantum expectation values at any temperature. The fact that Phase I adopts a close-packed hcp structure supports this.

Comment 2: *It is mentioned briefly that "Our potential does not allow for bond breaking" (p. 259), which appears to be strange to me. If the ML force field is able to reproduce a DFT energy landscape, then it should be able to break bonds as well (or badly) as DFT describes it. This would be relevant for fully describing the phase diagram, and one would hope to read at least a thorough justification why the bond breaking could not be achieved despite the authors' best efforts and why this does not affect any of the other results.*

Response: The functional form of our potential is based around molecules rather than atoms. A potential which describes the liquid-liquid transition is highly desirable, but would have to be built in a different, non-hierarchical way based on atoms, not molecules. Such a potential could not be made using the current formalism. A positive aspect of using molecular potentials (as opposed to atomic potentials) is that it shows that the densification of the liquid is not due to molecular dissociation. To emphasize all this, we added the word "molecular" to the title.

Comment 3: *Minor aspects of presentation: It seems strange to state that "Most information is gleaned from spectroscopy" (l. 34) and then to cite two papers which are strongly reliant on diffraction rather than spectroscopy. In Fig. 3a, it is not explained what the dashed line represents and the labels of the inset are*

very difficult to read. Aspects of the method such as "the projection method" for Raman predictions (l. 359) should be explained in more detail, so that readers can better understand what is being done here.

Response: We split the experimental references across two sentences, and to avoid confusion removed the reference to spectroscopy in the "X-ray sentence". The dashed line in Fig 3a is the EOS (i.e., V-P curve) of liquid H₂; In addition, we added a reference for the projection method and a short explanation.

Comment 4: *Data/Methods availability: I do not find the data availability statement that "All relevant data are available from the authors" (l. 388) to be fully satisfactory. In fact, it is not easily possible to understand from the Methods section how exactly the potential was fitted, much less to completely reproduce the authors' work. This is important especially given the apparent combination of classical and ML fitting methods (which is interesting!). The very terse presentation in the Methods section also leads to open questions on the side of the reader, e.g. when "the short-ranged Coulomb and van der Waals potentials" (l. 311) are mentioned - surely both these types of interaction are long-ranged? In summary, full details of the method should be given (those parts that do not fit within the length limit of the Methods could be placed in the SI), and it should be clarified whether the LAMMPS implementation will be published, and when, because this would strongly increase the usefulness of the authors' work to other researchers.*

Response: The DFT trajectory data used in fitting comprise hundreds of GB, and some of them were also used in other studies. We will make the relevant snapshots used in the fitting available via the Edinburgh University datashare repository system. We will make the LAMMPS implementation (c++ code) available as supplemental material

Detailed responses to Reviewer 3

General Comments: *The manuscript reports the results of MD simulations of molecular hydrogen with a potential fitted to DFT energies. They explain many of the unusual properties of hydrogen at high pressures. I think the research is a valuable contribution and should be published after the authors provide more details about their calculation.*

Response: We really appreciate the positive and very encouraging comments by the reviewer. With regard to the reproducibility, we have provided all the AIMD dataset, machine learning parameters and

the code for LAMMPS runs in this resubmission.

Comment 1: *The potential was constructed using the PBE density functional. Can the authors justify the use of this functional? Note PBE does not include the van der Waals interaction and has been shown to be less accurate for hydrogen than those functionals which do have VDW interactions. Perhaps line 99 "long-range interactions" should be clarified.*

Response: The PBE functional has become the de facto standard in studies of molecular systems, and high pressure hydrogen. PBE has been criticised for overstabilising the metallic phases, however in this work we only consider molecular phases, so this is not a concern. In our 2017 paper: "The role of van der Waals and exchange interactions in high-pressure solid hydrogen S Azadi, GJ Ackland PCCP 19, 21829" we showed that the various functionals including vdW corrections give just as large a spread of results as those without.

As further evidence, we recalculated the energies from our database of configurations using the vdW-DF proposed by Dion et al (implemented as optPBE-vdW in VASP). As the attached figure shows, the vdW term makes very little difference apart from a systematic shift in the energy due to using the free-atom as a reference state.

Figure R2 | Comparison of energies of fitted configurations, from MD at various pressures, with and without vdW corrections, now added in Supplementary materials

Comment 2: *I am concerned that the study is not reproducible. Neither the detailed procedures of fitting the DFT potentials nor the results of fitting are available. I do not regard communication with the authors as suitable alternative.*

-In doing the fit what is the objective function? energies or forces or both?

-what are the calculated values of the Morse potential?

-what are the parameters in Eq 3 for the non-bonded potential?

-Provide details of the Kernel Ridge regression?

-How many configurations are in the training set?

Response: We now provide a more detail of the machine learning model and parameterization in the last submission.

1) In our machine learning model, the objective function is energies and atomic forces. In order to simulate large systems, the total energy is expressed as a linear combination of the sum of local energy contributions from all the atoms. In this scenario, each atomic energy or force contribution depends only on its local environment, which is represented by a feature space vector or fingerprint so as to make the problem more amenable to a machine-learning representation.

2) The bonding energy in the Morse model is expressed as $D_e \cdot (e^{-2\alpha(r-r_e)} - 2e^{-\alpha(r-r_e)})$, where $D_e = 13.611$ eV/Å, $\alpha = 2.4352$ Å⁻¹, $r_e = 0.7365$ Å. Beside, we also provide the parameters for the Harmonic term (given by $0.5K \cdot (r-r_e)^2$, $K = 33.86$ eV/Å, $r_e = 0.730$ Å), which can prevent artificial bond-breaking events at high pressure (>120 GPa) during the long-time MD simulations.

3)The fingerprints for pairwise interactions are built using Gaussians with a smooth cutoff in the form

$$V_i^k = \sum \exp(-|r_{ij}/\eta_k|^2) f_{cut}(r_{ij}) \quad (1)$$

which are combined with damped sinusoidal functions in the form

$$V_i^k = \sum \sin(kr_{ij}) \exp(-r_{ij}/\eta_k) f_{cut}(r_{ij}) \quad (2)$$

where r_{ij} is the distance between atom i and j , and k is assigned as integers from the 1 to 8, $\eta_k = \eta_0 1.28^{k-1}$ with $\eta_0 = 0.6$. $f_{cut}(r_{ij}) = 0.5[1 + \cos(\pi r_{ij}/R_c)]$ is a damping function for atoms within the cutoff distance $R_c = 6.5$ Å. The parameters and LAMMPS-compatible code implementation are given in the supplementary information.

4) In the present work, the relationship between fingerprint vector and target (energy and atomic forces) is mapped by the kernel ridge regression (KRR) method (Kung, S. Y. Kernel Methods and

Machine Learning. (Cambridge University Press, 2014)). To be specific, the mapping function is given by a linear combination of kernel functions:

$$E_i = \sum W_t K(V_i, V_t) + b_0 \quad (3)$$

Here, K is a linear kernel function, whereas W_t and b_0 denote the weighting coefficient and a constant obtained from the fitting procedure, respectively. t labels each reference atomic environment and V_t is its corresponding fingerprint vector. For the local energy corresponding to pairwise interaction we adopt a linear kernel function of the form $K(x, y) = x \cdot y$, while for the orientation-dependent component, a RBF kernel function of the form $K(x, y) = \exp(-\sigma|x - y|^2)$ is used, where σ is length-scale parameter ($\sigma = 0.001$ in this work).

5) The final accumulated training dataset includes up to 41468 configurations, which are provided as a separate file in the Supplementary Information.

We have clarified these in the revised manuscript as part of the Supplementary Information. Besides, we have provided the complete AIMD dataset, machine learning parameters and the C++ code for LAMMPS implementation.

Comment 3: *On line 305 It is mentioned that "rigorous testing in MD is essential" Can the authors be more precise?*

Response: We refer to the fact that in MD the system is completely free to find any low energy configuration - in particular those which are high energy in DFT and therefore not in the fitting database. We have a long experience in potential-making, and have often seen papers in which potentials are published after testing against an array of static configurations but never used for MD. Often, these potentials are never used again and one eventually learns of their instability through personal communications.

Comment 4: *How good is the fit to DFT forces? (e.g. the rms value of the error in atomic units) Using this value another author could compare their ML potential.*

Response: We thank the reviewer for this important suggestion. We have followed this suggestion by adding the error evaluation, as shown in Fig. R3. The low mean absolute error (MAE) for both potential energy (MAE = 0.725 meV/atom) and atomic force (MAE = 0.102 eV/Å) indicates a very high fidelity

of our HMLP compared to the DFT counterpart.

We have clarified this in the revised manuscript.

Figure R3 | Performance of our ML potential compared with the AIMD references. a. Per-atom potential energy of all configurations in the training dataset. **b.** Atomic force for randomly selected 300 configurations in the training dataset. A perfect correlation with the DFT values would correspond to the red lines. MAE represents mean absolute error.

In summary, we express our deep appreciation to all the three reviewers again for their careful reviews of this manuscript and their various constructive comments and suggestions. We have incorporated their suggestions into the revised manuscript as appropriate, and hope the improved paper is now acceptable for publication in Nature Communications.

REVIEWER COMMENTS

Reviewer #1 (Remarks to the Author):

The manuscript has been considerably improved and the presented evidence for structural changes underlying the turnover of the melting line is much more convincing than the one in the original version. I also appreciate the longer MD simulations performed for larger system sizes. I still do not understand why in Fig.1 there are for each pressure value two red circles representing phase II. The text commenting Fig.1 on p.5 instead mentions orange hexagon symbols and red triangles while no such symbols seem to be present in the figure. The authors should make sure that the figure is consistent with the caption and with the description in the text. After fixing these issues I recommend the manuscript for publication in Nat. Comm.

Reviewer #2 (Remarks to the Author):

The authors have overall made a satisfactory response to the referee's comments, which includes more extended MD simulations (to provide robustness for the phase diagram prediction) and a new way of determining structural properties (using one of the "fingerprints" from the potential fitting is a nice idea). However, I feel that a number of open questions remain.

(1) In their response to referee 2 / comment 2 (#2.2), the authors justify their development of a molecular (rather than atomic) potential and this is indeed something which makes the paper different from other ML potentials in the literature. However, I am not convinced about the authors' argument that "A positive aspect of using molecular potentials (as opposed to atomic potentials) is that it shows that the densification of the liquid is not due to molecular dissociation" (p. 6 of the rebuttal). The present simulations only show that a densification is observed within the constraints imposed by limiting the potential to molecular units, but they do not allow to rule out the possibility of atomic dissociation, which could, independently, lead to densification.

(2) More details have been provided in the SI (following comments #2.4 and #3.2) although there are several aspects which are not yet explained. Specifically:

- "the sequential feature selection algorithm is adopted to select the best 16 function forms" (p. 2 of the SI) - details of this selection algorithm should be given, and it should be explained how many possible features there are overall and how strongly the results depend on this choice.

- "All the parameters are determined during the training process, with the help of cross-validation and regularization methods." (p. 3) - these are presumably very important aspects of the method. What is the number of parameters? How is the cross-validation done and what are the numerical results? What is the purpose of "regularization methods" and what are the details of those?

- "A similar function form to Eq. (3)" (p. 3) - the exact form should be specified, and the choice of sigma in the following sentence should be explained.

Overall, I feel that the construction of the hierarchical ML model is one of the interesting aspects of the work (and one that will determine how interesting it will be to a broad audience). I think the manuscript would therefore benefit from a more detailed discussion, e.g. showing the error of the model after each "hierarchical" step and how the next additive term (covalent, nonbonding, orientation-dependent) improves this error.

(3) I think comment #3.3 ("It is mentioned that "rigorous testing in MD is essential" Can the authors be more precise?") is an important point, and even though the authors provide a general response in their rebuttal, I would have hoped to see some changes being made to the manuscript or SI in response, rather than only making general comments in the rebuttal letter. Can the authors provide some details of "rigorous testing in MD", e.g. how the success (or otherwise) of an earlier potential was determined? At the moment, the paragraph at the top of p. 22 is not supported by actual results, and giving even a few selected examples would make it much stronger.

(4) The implementation of the method is now provided as a single .gz compressed text file. This seems to meet the minimum requirement for reproducibility, although those files could be provided in a much more "user-friendly" way: e.g., an example calculation is mentioned ("Run the example which includes test input file and H-H pair-bond file", l. 4889) but in the current form, the reader would have to search and copy/paste the required information from different parts of the text file. This may seem like nitpicking, but I think it would help readers to actually use the authors' method.

Reviewer #3 (Remarks to the Author):

The authors still did not elaborate on what the phrase "rigorous testing of the potential in MD is essential" means in practice.

Responses to Reviewers' Reports on NCOMMS-20-14026A

We thank all the three referees again for their careful review of this manuscript. In the last submission, all the three reviewers were largely supportive of this paper for publication in Nature Communications, aside from a few minor questions or issues to be further clarified. As detailed below, we have adequately addressed these questions or issues raised by the three reviewers, and have revised the paper accordingly.

Reviewer 2 had several nice suggestions for a better presentation of our machine learning model, and we have followed these suggestions by explaining the algorithm and terminology thoroughly in the revised manuscript. Also a more friendly ReadMe text file is updated in our code package.

With these responses and improvements, we hope the paper is now acceptable for publication in Nature Communications.

Detailed responses to Reviewer 1

Comment 1: *The manuscript has been considerably improved and the presented evidence for structural changes underlying the turnover of the melting line is much more convincing than the one in the original version. I also appreciate the longer MD simulations performed for larger system sizes. I still do not understand why in Fig.1 there are for each pressure value two red circles representing phase II. The text commenting Fig.1 on p.5 instead mentions orange hexagon symbols and red triangles while no such symbols seem to be present in the figure. The authors should make sure that the figure is consistent with the caption and with the description in the text. After fixing these issues I recommend the manuscript for publication in Nat. Comm..*

Response: The issues with Fig 1 were a mistake on our part. We thank the reviewer for noticing this and have changed it to the correct figure and caption. All the data points are from the new, larger simulations: previous, shorter runs within the Phase I region are consistent with the phase diagram, but now omitted

Detailed responses to Reviewer 2

General Comment: *The authors have overall made a satisfactory response to the referee's comments, which includes more extended MD simulations (to provide robustness for the phase diagram prediction)*

and a new way of determining structural properties (using one of the "fingerprints" from the potential fitting is a nice idea). However, I feel that a number of open questions remain.

Response: We really appreciate the positive and very encouraging comments by the reviewer. With regard to the machine learning model, we address the range of technical comments quite thoroughly in our response below, and we believe that these responses should fully satisfy the reviewer.

Comment 1: *In their response to referee 2 / comment 2 (2.2), the authors justify their development of a molecular (rather than atomic) potential and this is indeed something which makes the paper different from other ML potentials in the literature. However, I am not convinced about the authors' argument that "A positive aspect of using molecular potentials (as opposed to atomic potentials) is that it shows that the densification of the liquid is not due to molecular dissociation" (p. 6 of the rebuttal). The present simulations only show that a densification is observed within the constraints imposed by limiting the potential to molecular units, but they do not allow to rule out the possibility of atomic dissociation, which could, independently, lead to densification.*

Response: We agree that dissociation certainly does lead to densification, and drives the liquid-liquid phase transition. What we showed is that a) in our simulations the densification of the liquid is not due to molecular dissociation; b) atomic dissociation is not necessary for densification in the real system.

Since the unclear statement was only in the response, no change was made to the paper.

Comment 2: *More details have been provided in the SI (following comments 2.4 and 3.2) although there are several aspects which are not yet explained. Specifically:*

- "the sequential feature selection algorithm is adopted to select the best 16 function forms" (p. 2 of the SI) - details of this selection algorithm should be given, and it should be explained how many possible features there are overall and how strongly the results depend on this choice.

Response: The technical details of the fitting have been include in the Supplementary Information.

In the present work, 96 possible features are built firstly based on J. Phys. B: Atom. Molec. Phys. **11**, 633 (1978), in which A. Koide proposed a model for the angular-dependent interaction between two linear molecules.

The motivation of using the sequential feature selection (SFS) algorithm is to improve the compu-

tational efficiency of our potential and reduce the generalization error of the model by removing less relevant features or noise for our solid H₂ system.

The detail of the selection algorithm is addressed in Fig. R1. It is outlined in pseudo code below:

Input: the set of all 96 features, $Y = \{y_1, y_2, \dots, y_{96}\}$

Output: a subset of features, $X_k = \{x_j | j = 1, 2, \dots, k; x_j \in Y\}$

Initialization: $X_0 = Y, k = 96$

Step 1 (Conditional Exclusion):

Use greedy search algorithms to find three features from our feature subset, X_k , and remove them to if they can improve the fitting precision; 2) $k = k - 3$; 3) Go to Step 2;

Step 2 (Conditional Inclusion):

Search for two features (from the removed ones) that improve the performance of the ML potential if they are added back to the feature subset, X_k ; 2) $k = k + 2$; 3) Go to Step 1;

Termination: Stop when the number of features in X_k equals 16.

Figure R1 | Flow chart of the sequential feature selection (SFS) algorithm used in the present work. The SFS algorithm helps us select 16 best features from the 96 candidates. The details of the SFS algorithm is illustrated in Supplementary Note 3.

- "All the parameters are determined during the training process, with the help of cross-validation and regularization methods." (p. 3) - these are presumably very important aspects of the method. What is the number of parameters? How is the cross-validation done and what are the numerical results? What is the purpose of "regularization methods" and what are the details of those?

Response: We thank the reviewer for raising this point. We could indeed have been clearer about this.

1. The parameters of our machine learning model are listed in the parameter file of Para_ML_pot.txt, which has been provided in the supplementary package of HMLP_code.tar.gz.
2. As for the cross-validation, our original training dataset was randomly partitioned into 5 equal sized subsamples. Of the 5 subsets, a single subset was retained as the validation data for testing the model, and the remaining 4 subsets are used as training data. The cross-validation process was then repeated 5 times, with each of the 5 subsets used exactly once as the validation data. The 5 results were then averaged to estimate the fitting error. In our work, the cross-validation gives rise to a mean absolute error of 0.007649 eV/atom eventually, indicating a high fidelity and transferability of our interatomic potential.
3. We use L2 regularization method (or Ridge Regression), which is a technique to improve the generalizability of our machine learning model by avoiding overfitting. It adds “squared magnitude” of fitting parameters as penalty term to the loss function, that is

$$Loss = \sum_i (y_i - f_i(X_i, \beta_0, \beta_1, \dots, \beta_p))^2 + \lambda \sum_{j=1}^p \beta_j^2 \quad (1)$$

Where y_i is the target value, X_i is the fingerprint vector, β_j is the fitting parameters and λ is the weight factor. In the present work, $\lambda = 10^{-5}$ with the best prediction ability.

- ”A similar function form to Eq. (3)” (p. 3) - the exact form should be specified, and the choice of sigma in the following sentence should be explained.

Response: We thank the reviewer for this suggestion.

1. For the orientation-dependent component, the mapping function has the function form as following:

$$E_i^{orient} = \sum_t W_t K(V_i, V_t) + b_1 \quad (2)$$

$K(x, y) = \exp(-\sigma|x - y|^2)$ is used, while W_t is the weighting coefficient and b_1 is a constant.

2. σ is selected so that the machine learning model can possess the smallest fitting error.

In summary, we have clarified the details of relevant data mining techniques or terminologies in the revised manuscript and SI. Also, two handbooks of machine learning are cited in order to help readers understand these terminologies.

[1] Raschka, S. Python Machine Learning (Packt Publishing, Birmingham, 2015).

[2] Rostamizadeh, A., Talwalkar, A., Mohri, M. Foundations of Machine Learning (MIT Press, London, 2012).

Overall, I feel that the construction of the hierarchical ML model is one of the interesting aspects of the work (and one that will determine how interesting it will be to a broad audience). I think the manuscript would therefore benefit from a more detailed discussion, e.g. showing the error of the model after each "hierarchical" step and how the next additive term (covalent, nonbonding, orientation-dependent) improves this error.

Response: We thank the reviewer for this important suggestion.

As the reviewer suggested, we evaluate the performance of our ML model after each "hierarchical" step. As shown in Fig. R2, correlation plots of atomic force and potential energy make this clear: the fitting error decrease sharply when we add the next additive term.

We have added this discussion to the revised manuscript.

Figure R2 | Performance of our ML potential after each hierarchical step **a.** atom force from covalent term. **b.** ML atomic force with or without the orientation-dependent contribution. **c.** Scatter plots for per-atom potential energy with or without the orientation-dependent contribution. A perfect correlation with the DFT values would correspond to the red lines. **d.** Comparison of the mean absolute error (MEA) after different hierarchical step.

Comment 3: *I think comment 3.3 ("It is mentioned that "rigorous testing in MD is essential" Can the authors be more precise?") is an important point, and even though the authors provide a general response in their rebuttal, I would have hoped to see some changes being made to the manuscript or SI in response, rather than only making general comments in the rebuttal letter. Can the authors provide some details of "rigorous testing in MD", e.g. how the success (or otherwise) of an earlier potential was determined? At the moment, the paragraph at the top of p. 22 is not supported by actual results, and giving even a few selected examples would make it much stronger.*

Response: We do rigorous testing in MD as follows: Firstly, we test each iteration of the potential by running MD simulations in the NPT ensemble, cycling the pressure and temperature. When an unexpected phase was found, the trajectories of this unexpected phase were calculated using DFT, then added to the training set and the potential was re-fit.

Here, we show an example how the database is updated by the iterative scheme. Using an earlier trial HMLP, the MD simulation at 100 GPa and 50 K leads to a wrong phase structure, which was found to be unstable in DFT. Fig. R3a and 3b show the structures from the MD simulation after two successive iterations. However, upon further iteration, we successfully got the $P2_1/c-24$ structure when the MD simulations are re-run with the re-fitted potential, as shown in Fig. R3c.

Figure R3 | Structures from MD simulation tests in our iterative process a and b shows unexpected structures from the MD simulation with earlier potential at pressure of 100GPa and temperature of 50 K. c. $P2_1/c-24$ structure from the MD simulation with updated potential.

Comment 4: *The implementation of the method is now provided as a single .gz compressed text file. This seems to meet the minimum requirement for reproducibility, although those files could be provided in a much more "user-friendly" way: e.g., an example calculation is mentioned ("Run the example which includes test input file and H-H pair-bond file", l. 4889) but in the current form, the reader would have to search and copy/paste the required information from different parts of the text file. This may seem like nitpicking, but I think it would help readers to actually use the authors' method.*

Response: We are keen that our potential should be as user-friendly as possible. The code was submitted as a gzipped tar archive, which should have consisted of multiple separate files rather than a single text file. We are sorry about the difficulties experienced by the reviewer in viewing our code and example input files. On this resubmission, we have double-checked the .tar.gz file to ensure that it is indeed multiple files. The .tar.gz archive consists of 3 directories which contain the following:

- doc – a documentation page explaining the LAMMPS implementation of the potential.
- example – sample LAMMPS input files (4 files) which use the ML potential, ready to be run by the user. The in.lammps file has been updated to be more descriptive and more user-friendly.
- HMLP code – source code for the HMLP LAMMPS plugin. This has been updated to include explicit instructions on how to compile LAMMPS + HMLP using two different methods.

Detailed responses to Reviewer 3

Comment 1: *The authors still did not elaborate on what the phrase "rigorous testing of the potential in MD is essential" means in practice.*

Response: In the revised manuscript, we clarified the sentence... "A good fit to DFT energies of known phases is not evidence that other phases are unstable: rigorous testing of the potential in MD is essential." To... "A good fit to DFT energies of known phases is not evidence that other phases are unstable: we test each iteration of the potential by running MD simulations in NPT ensemble, cycling the pressure and temperature. When an unexpected phase was found, this was calculated using DFT, and if poorly described, it was added to the training set and the training redone".

In summary, we express our deep appreciation to all the three reviewers again for their reviews of this manuscript as well as their various constructive comments and suggestions. We have incorporated their suggestions into the revised manuscript as appropriate, and hope the improved paper is now acceptable for publication in Nature Communications

REVIEWERS' COMMENTS

Reviewer #2 (Remarks to the Author):

The authors have addressed the referees' concerns in their rebuttal. I feel that more detail would still help the paper, but these are mainly issues of presentation.

Specific comments:

- On p. 22, a minimal description of the iteration procedure is now given. I think the paper would benefit very strongly from providing more information: how do the authors define "an unexpected phase" (by manual inspection?); how do they know it is "poorly described"? What are the values for "cycling the pressure and temperature" and how are they chosen? Giving more details will help the readers to appreciate the authors' work, and ideally to build on it in future studies.

- In the new Fig. 1, the meaning of the different symbols is unclear. Please add a description to the legend to make the presentation self-contained.

- I think that the new Fig. S3 is a useful addition, illustrating the hierarchical fitting (which is one of the key aspects of the paper), and that this figure could in fact be moved to the main text if space allows. In any case, the legend should be improved by explaining in more detail what is shown in panel a ("atom force from covalent term"), and correcting "MEA" to "MAE".

- "the cross-validation gives rise to a mean absolute error of 0.007649 eV/atom, indicating a high fidelity and transferability of our interatomic potential" (Supplementary Note 3). Is it justified to give this number of significant digits, especially given the inherent error of the DFT data (which, in fact, should be discussed here)? Also, I am not sure whether the claim about transferability is justified (if the cross-validation is done for a dataset that is limited to a number of phases, this does not prove transferability to a completely different phase).

Responses to Reviewer's Reports on NCOMMS-20-14026B

Detailed responses to Reviewer 2

General Comment: *The authors have addressed the referees' concerns in their rebuttal. I feel that more detail would still help the paper, but these are mainly issues of presentation. Specific comments:*

Comment 1: *- On p. 22, a minimal description of the iteration procedure is now given. I think the paper would benefit very strongly from providing more information: how do the authors define "an unexpected phase" (by manual inspection?); how do they know it is "poorly described"? What are the values for "cycling the pressure and temperature" and how are they chosen? Giving more details will help the readers to appreciate the authors' work, and ideally to build on it in future studies.*

Response: We have re-written the iteration procedure in the revised manuscript as following:

"we test each iteration of the potential by running MD simulations in NPT ensemble, cycling the pressure and temperature between the regions expected for Phases I, II and III to ensure that all phase transitions were between the fitted phases. When a crystal structure different from the fitted one was found, this new phase was calculated using DFT. If this showed that the ML potential described the phase poorly, it was added to the training set and the training redone."

Comment 1: *- In the new Fig. 1, the meaning of the different symbols is unclear. Please add a description to the legend to make the presentation self-contained.*

Response: We thanks the reviewer for this suggestion.

We have clarified this the revised manuscript as following:

"Blue squares represent the melting points form Z-method while the data of stable Phase I and Phase II appears as orange hexagon and red triangles, respectively."

Comment 3: *- I think that the new Fig. S3 is a useful addition, illustrating the hierarchical fitting (which is one of the key aspects of the paper), and that this figure could in fact be moved to the main text if space allows. In any case, the legend should be improved by explaining in more detail what is shown in panel a ("atom force from covalent term"), and correcting "MEA" to "MAE"*

Response: We thank the reviewer for the suggestions.

Here "atom force from covalent term" refers to the covalent term only, with no intermolecular interactions. The DFT data used to fit this term is $(F_1^{\mu_{12}} - F_2^{\mu_{12}})/2$, where $F_1^{\mu_{12}}$ is the component of atomic force projected down the molecular axis.

We have followed these suggestions in the revised manuscript. We prefer not to include the figure in the main paper, because the details of the figure depend on the DFT configurations actually used. Consequently, we think that this figure will mainly be of interest to specialists.

Comment 4: - *"the cross-validation gives rise to a mean absolute error of 0.007649 eV/atom, indicating a high fidelity and transferability of our interatomic potential" (Supplementary Note 3). Is it justified to give this number of significant digits, especially given the inherent error of the DFT data (which, in fact, should be discussed here)? Also, I am not sure whether the claim about transferability is justified (if the cross-validation is done for a dataset that is limited to a number of phases, this does not prove transferability to a completely different phase).*

Response: We thank the reviewer for raising this issue.

The method of cross-validation is used to evaluate the difference between the target within the training/reference data (DFT data) and ML prediction. The inherent error of the reference data is not considered here.

As for the transferability, it should work among the structure of Phases I, II, III and liquid (below 160 GPa) since our test data are randomly selected from the structures of these phases. However, a further transferability to a completely different phase (at higher pressures) is beyond the scope of the present HMLP. We expect that this work will stimulate further study of phase transition in solid H₂ of higher pressure region.

We have clarified this the revised manuscript.